# Development and validation of a pharmaceutical assessment screening tool to prioritise patient care in a tertiary care hospital

Cheok Ee Chang[1]*, Rahela Ambaras Khan[1], Chan Yen Tay[1], Baavaanii Thangaiyah[1], Victor Sheng Teck Ong[1], Sabariah Pakeer Oothuman[1], Shazwani Zulkifli[1], Nur Fatin Najwa Azemi[1], Pavithira Subramaniam[2]

1 Pharmacy Department, Hospital Kuala Lumpur, Ministry of Health, Kuala Lumpur, Malaysia, 2 Pharmacy Department, Hospital Tunku Azizah, Ministry of Health, Kuala Lumpur, Malaysia

* carolce8@gmail.com

## Abstract

### Background

Clinical pharmacy plays an integral role in optimizing inpatient care. Nevertheless, prioritising patient care remains a critical challenge for pharmacists in a hectic medical ward. In Malaysia, clinical pharmacy practice has a paucity of standardized tools to prioritise patient care.

### Aim

Our aim is to develop and validate a pharmaceutical assessment screening tool (PAST) to guide medical ward pharmacists in our local hospitals to effectively prioritise patient care.

### Method

This study involved 2 major phases; (1) development of PAST through literature review and group discussion, (2) validation of PAST using a three-round Delphi survey. Twenty-four experts were invited by email to participate in the Delphi survey. In each round, experts were required to rate the relevance and completeness of PAST criteria and were given chance for open feedback. The 75% consensus benchmark was set and criteria with achieved consensus were retained in PAST. Experts' suggestions were considered and added into PAST for rating. After each round, experts were provided with anonymised feedback and results from the previous round.

### Results

Three Delphi rounds resulted in the final tool (rearranged as mnemonic 'STORIMAP'). STORIMAP consists of 8 main criteria with 29 subcomponents. Marks are allocated for each criteria in STORIMAP which can be combined to a total of 15 marks. Patient acuity level is determined based on the final score and clerking priority is assigned accordingly.

**Data Availability Statement:** Minimal data set can be accessed from 10.6084/m9.figshare.22121492.

**Funding:** The authors received no specific funding for this work.

**Competing interests:** The authors have declared that no competing interests exist.

## Conclusion

STORIMAP potentially serves as a useful tool to guide medical ward pharmacists to prioritise patients effectively, hence establishing acuity-based pharmaceutical care.

## Introduction

Clinical pharmacy services have commenced in many government hospitals in Malaysia since the year 2000. The services provided include medication counselling, medication reconciliation, bedside dispensing, therapeutic drug monitoring (TDM), and many others [1]. Clinical pharmacists participate in daily ward rounds with a multidisciplinary team, providing pharmaceutical care to ensure optimal medical treatment plans for patients.

Clinical pharmacists also contribute to identifying and reducing drug-related problems. A local pharmaco-economic study done in 2001 at an intensive care unit (ICU) in Penang General Hospital shows that 95% of the interventions done by clinical pharmacists were accepted by physicians, resulting in a cost-saving worth of RM 15,227 for drug expenses [2]. Besides, a local study done by Fahrni et al. on the elderly population shows that the prevalence of inappropriate prescribing in the elderly was 58.5%, and with every increase in the number of medications prescribed, the likelihood of potentially inappropriate medications increased by 20% [3]. With clinical pharmacy service, drug-related issues could be addressed earlier through pharmaceutical care interventions, reducing the risk of adverse drug events.

To ensure the quality of pharmaceutical care provided by ward pharmacists, the Pharmaceutical Services Programme under the Ministry of Health Malaysia has developed a standardized clinical pharmacotherapy review form (CP2) for case clerking purposes [4]. CP2 is an important documentation for clinical pharmacists to monitor patients' progress and rationalise the therapy and dose regimen with patients' characteristics, electrolytes, organ functions, and other relevant laboratory and radiological findings. Any pharmaceutical care issue (PCI) identified will be pointed out to the prescribers and documented in the CP2 forms. The CP2 will be transferred to another pharmacist alongside with patient transfer to facilitate continuous monitoring for optimal pharmaceutical care. Despite the comprehensive form for monitoring patients' progress, clinical pharmacists have little guidance on how to select the case to be clerked in medical wards considering the number of admissions in the busy medical wards.

Prioritising patients to optimise the delivery of pharmaceutical care is gaining more attention as patients are becoming more complex, with multiple co-morbidities and prescribed with multiple medications. Clinical pharmacists in Hospital Kuala Lumpur (HKL) often face challenges in prioritising patient care when they are located in a highly active medical ward, with an approximate ratio of 1 pharmacist to 30 patients. Within limited working hours, pharmacists need to multitask to fulfil daily duties, which include reviewing all patients' medications and prioritising patients for case clerking, alongside other roles such as managing drug-related issues, drug counselling, bedside dispensing and more. Pharmacists without much clinical experience may find it difficult to prioritise patient care in the ward. Without proper guidance, they may resort to work in response to urgent demands only [5]. In such a way, patients of higher acuity may not receive adequate pharmacist reviews and both patient and organisational needs may not be achieved. Therefore, a pharmaceutical assessment screening tool is necessary, to assist clinical pharmacists in prioritising patient care while executing their duties in medical wards [5].

Several assessment tools have been designed and validated to assist the pharmacists in identifying patients with higher complexity requiring intensive pharmaceutical care [6–9].

Nonetheless, most of these screening tools are directly integrated into the respective countries' electronic healthcare systems, in which patients' risk factors are calculated electronically to give result in patient prioritisation. It is difficult to adopt these tools directly into Malaysian local hospitals which have heterogeneity of electronic and manual-based healthcare systems. To date, clinical pharmacy service in Malaysia faces a paucity of standardised patient screening tools to guide pharmacists in prioritising patient care. Thus, our study planned to adapt the available assessment tools and modify them based on the accessibility and feasibility of our local settings. We aim to develop and validate a pharmaceutical assessment screening tool (PAST) to assist clinical pharmacists working in the general medical wards in HKL to prioritise their patients and hence to clerk CP2 on patients of high acuity who necessitate more attention and more intensive care from the pharmacists.

## Methods

The pharmaceutical assessment screening tool (PAST) was developed through two major phases, namely: (1) Development of PAST; and (2) Validation of PAST via a three-round Delphi method. The study was approved by the Medical Research Ethics Committee (MREC), Ministry of Health, Malaysia (NMRR-21-27-57897) and the Clinical Research Committee of HKL. It was conducted in compliance with ethical principles outlined in the Declaration of Helsinki and Malaysian Good Clinical Practice (GCP) Guideline.

### Development of PAST

PAST is developed from published screening tools. The criteria selected for PAST are mainly adapted from the PAST tool designed by pharmacists in a UK hospital [6], the Assessment of Risk Tool (ART) by a New Zealand hospital [8], DART (Drug-Associated Risk Tool) tool [9], and UK NHS (National Health Service) Prevent Tool [10]. These tools were adapted to provide the basis for stratifying patients according to their pharmaceutical complexity and subsequently the patient acuity level to guide the pharmacists to prioritise patient care.

A clinical pharmacy taskforce of 9 members, comprising senior clinical pharmacists and other ward pharmacists, conducted a discussion on the content of PAST. The content of the tool was further modified to meet the needs of local medical ward settings. A scoring system was also created for the tool. Scores were assigned for each criteria in PAST to determine the overall patient acuity level (PAL). The initial weightage of scoring for each criteria was determined by the taskforce based on their clinical knowledge and experience. The most pharmaceutically complex patients are expected to receive a PAL score of at least 3, while the least complex case receives a PAL score of 0.

### Validation of PAST

The PAST validation process involved a three-round Delphi survey method. Three rounds were used as some experts reported that most changes are found in the first two rounds, and the rounds should be done as few as possible to avoid time pressure and respondent fatigue resulting in high expert attrition [11, 12].

Expert panels were selected from Hospital Kuala Lumpur (HKL), Hospital Tunku Azizah (HTA) and the Institute of Respiratory Medicines (IPR) as these three facilities are in nearby vicinity in the heart of Kuala Lumpur. The panels of experts were selected based on the following criteria:

1. Pharmacists who have Masters or PhD in Clinical Pharmacy; and

2. Senior pharmacists with working experience of 7 years or more in the pharmacy field (Pharmacy grade UF48 and above in government hospitals)

An invitation email to participate in the Delphi survey was sent out to the experts. Experts who agreed to participate signed a consent form for ethical consideration. The same group of experts who consented participated in all three rounds of the Delphi survey. We aimed for a response rate of at least 70% from the experts to maintain the rigour of the Delphi method [13].

Before the actual Delphi survey, a pilot Delphi round was conducted with 5 experts in early April 2021 to assess the face validity of the PAST criteria and the Delphi survey questions and allow opinions on the PAST. These 5 experts were not included in Delphi survey rounds 1–3. Suggestions given during the pilot Delphi round were discussed in the pharmacy taskforce to improve on the PAST before the actual Delphi survey.

The actual Delphi survey was conducted online using Google Forms. At the beginning of each round, experts were reminded of the objective of PAST is to assist the clinical pharmacists in the general medical wards to prioritise the patients for case clerking and pharmaceutical care. Hence, it is emphasized that the tool should: 1) focus on the most important factors in stratifying different patient acuity levels in general medical wards, 2) be easy to use in the time constraint of routine practice, and 3) be simple to be used for all pharmacists with different clinical experience or background. In each round, experts were given 2-week time to complete the questionnaire.

Delphi survey round 1 was conducted from 21 April 21 to 6 May 21. The PAST tool and its appendices together with the google form link to the Delphi questionnaire were emailed to the experts. Expert's feedback was then extracted from the Google Form website and analysed after 2 weeks.

Experts were asked to rate the relevance and completeness of each criteria in the PAST using the 5-point Likert scale: (5) strongly agreed; (4) in agreement; (3) neither in agreement nor in disagreement; (2) in disagreement and (1) strongly disagreed. Relevance is defined as the criteria that are essential and applicable for practice. Completeness is defined as the criteria that have all the necessary and appropriate information contained in the statement.

Each criteria was followed by two open-ended questions in the first round:

*1. Does the criteria above require any modification and addition in terms of content?*

*2. Does the criteria above require any modification in terms of sentence structure?*

For the second and third rounds of the Delphi survey, open-ended questions were simplified to a column with "General Comments" at the end of each criteria. This is to reduce confusion and allow more flexibility for experts to provide their opinions. At the end of the questionnaire, there will be another two open-ended questions:

*3. If you think that the draft PAST ignores some essential aspects in prioritizing a patient's acuity level in a general medical ward, please list out each item in detail.*

*4. Referring to the total scoring, do you agree with the scoring method?* (Followed by a dichotomous yes or no) *If you do not, please state your reason and suggestion.*

Fig 1 below summarizes the subsequent actions to be taken based on the data analysis. We first considered retaining or removing the criteria based on its "Relevance". If the criteria was retained, we then looked at its completeness to decide if we need to modify the criteria based on expert suggestions. In contrast, if a criteria does not reach consensus for its relevance, it will be removed from the list and will not be considered for its completeness.

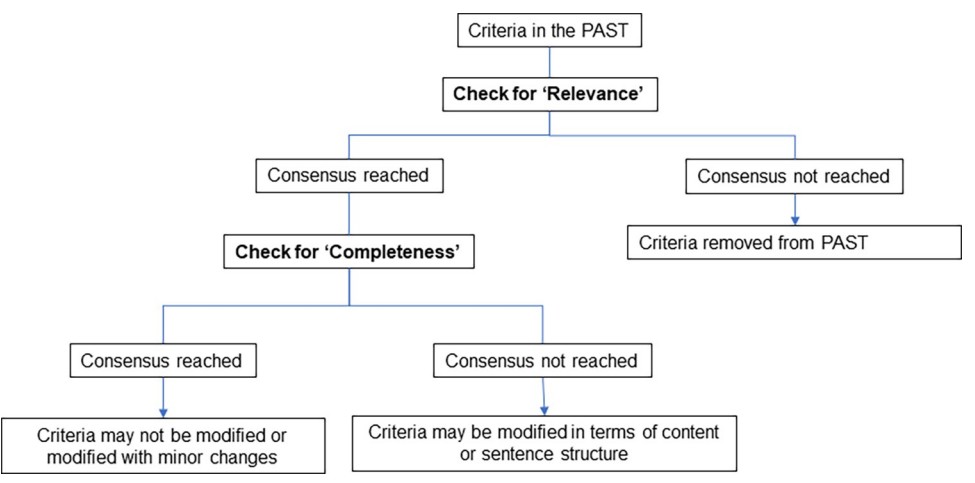

**Fig 1. Consensus of agreement and actions to be taken.**

Delphi survey round 2 was conducted from 2 to 15th July 2021. In the second round, the modified PAST, results of data analysis from round 1 and expert feedback were sent to the experts for review. Experts rated the modified or newly added criteria using a 5-point Likert scale. Experts were also required to answer a dichotomous "Yes" or "No" for the scoring system of PAST and were given chance to express their opinions in the 'general comment' column at the end of each criteria and the overall PAST.

The third round of the Delphi survey was conducted from 3rd to 17th September 2021. In this final round, the modified PAST together with the results of suggestions from the second round was presented to the experts. Experts were required to rate again the relevance and completeness of each criteria in the modified PAST. All criteria that have ≥ 75% consensus on "Relevance" and "Completeness" will be retained in PAST. No further amendments were made after the third round. Fig 2 shows the process of the three-round Delphi survey.

## Data analysis

Descriptive statistics (i.e., Median, interquartile range, frequencies and percentages) were used to analyse the data, using the Microsoft Excel version 2111. We set the benchmark of consensus at 75%, in which a criteria will be retained or may be unmodified in the PAST when 75% or more of experts rating 'agreed' and 'strongly agreed' for that criteria [14]. A consensus of 75% or more is chosen as Keeney et al. has considered 75% as the minimum level of consensus which appears to be a more robust cut-off for panel agreement [12].

## Results

### Demographic data

24 expert panels comprising senior pharmacists working in HKL, HTA and IPR have consented to the study. Expert demographic data is presented in Table 1. Majority of the experts have Master in Clinical Pharmacy (22/24) while 2 (2/24) have PhD in Clinical pharmacy. Most experts have 11–12 years (10/24, 41%) of working experience in Pharmacy. Also, more than 58% (14/24) of them have working experience of 8 years and above in clinical pharmacy or ward pharmacy unit.

**Step 1: Development of PAST**

Based on literature review and discussion within research group committee

**Step 2: Selection of Expert Panels**

- Expert panels were selected from HKL, HTA and IPR based on the selection criteria below:
    1. Pharmacists who have Masters/ PhD in Clinical Pharmacy; and
    2. Senior pharmacists with working experience of 7 years or more in the pharmacy field (with grade UF48 and above)
- 24 experts consented to participate in this study.

**Step 3: Pilot Round of Delphi (1st - 3rd April 2021)**

For the pilot round, 5 experts are invited to:
    1. Assess face validity of PAST and Delphi questionnaire
    2. Give recommendations for the improvement on the draft PAST

**Step 4: Delphi Survey Round 1 (21st April to 6th May 21)**

- Experts were asked to rate the relevance and completeness of each criteria in the PAST using the 5-point Likert scale. Experts were also given a chance to give opinions through the open-ended questions in the questionnaire
- After a 2-week period, questionnaire responses will be collected and analysed. Suggestions were compiled and listed out.
- Response rate: 24/24 (100%)

**Step 5: Delphi Survey Round 2 (2nd - 15th July 2021)**

- Results of first round were sent to experts, together with:
    1. Criteria that have reached consensus and retained in PAST
    2. Criteria that were newly added or modified for rating
    3. Experts feedback from round 1
- Experts rated the relevance and completeness using the 5-point Likert scale and could give opinions in "General comments" section.
- Response rate: 23/24 (95.8 %)

**Step 6: Delphi Survey Round 3 (3rd – 17th September 2021)**

- Results of second round were sent to experts, together with:
    1. Criteria that have reached consensus and retained in PAST
    2. Criteria that were newly added or modified for rating
    3. Experts feedback from round 2
- Experts rated again relevance and completeness of each criteria that was modified
- Criteria that have reached consensus in Round 2 with no modifications was remained and no rating required.
- Response rate: 24/24 (100%)
- All items that have achieved consensus in both Relevance and completeness in Round 3 were retained in the final PAST with no modifications done.

**Fig 2. Flow chart of the three-round Delphi survey.**

## Delphi round 1

First Delphi round were responded by all 24 experts and the results are presented in Table 2. 5 out of 7 main criteria in PAST have achieved consensus in both relevance and completeness.

Table 1. Characteristics of expert panels.

| Demographics | | Pharmacist (n = 24) | |
| --- | --- | --- | --- |
| | | n | % |
| Gender | Female | 20 | 83.3 |
| | Male | 4 | 16.7 |
| Age | 30–34 | 5 | 20.8 |
| | 35–39 | 16 | 66.7 |
| | 40–45 | 3 | 12.5 |
| Qualification | Master | 22 | 91.7 |
| | PhD | 2 | 8.3 |
| Years of experience in Pharmacy (Years) | 9–10 | 6 | 25.0 |
| | 11–12 | 10 | 41.7 |
| | 13–14 | 5 | 20.8 |
| | $\geq 14$ | 3 | 12.5 |
| Years of experience of working in Clinical Pharmacy unit (Years) | $\leq 5$ | 3 | 12.5 |
| | 6–7 | 7 | 29.2 |
| | 8–9 | 8 | 33.3 |
| | $\geq 10$ | 6 | 25.0 |

The statement for "High Alert Medications" (HAM) attained consensus in relevance but not in completeness (70.8%), indicating insufficient information in the statement. There were experts suggesting to add on a time frame to patient on HAMs "for more than 24 hours", as well as suggesting to "narrow down" HAM list to medications that are clinically relevant in medical wards. Suggestions were considered and HAM list was narrowed down to contain only medically relevant HAM.

The criteria of "JKUT (Drug Review & Therapeutic Committee) Quota medication" did not achieve agreement among the experts (Relevance 50%, Completeness 54.1%). A few experts have raised opinions that medicines that are listed as "JKUT quota" in hospitals are often due to their costs but not to their clinical importance in pharmacological therapy, which

Table 2. Delphi round 1 results.

| | Criteria | | Median | IQR | Consensus % |
| --- | --- | --- | --- | --- | --- |
| | **Experts response rate (%)** | | 24/24 (100%) | | |
| H | **High Alert Medication**—Patient requiring High Alert Medications (HAM) | R | 5 | 1 | 87.5 |
| | | C | 4 | 2 | 70.8 |
| O | **Organ Dysfunction**—Presence of altered organ function of at least one organ in an acutely ill patient such that homeostasis cannot be maintained without intervention | R | 5 | 0.25 | 100 |
| | | C | 5 | 1 | 87.5 |
| T | **Therapeutic Drug Monitoring (TDM)**—Medication(s) that has a narrow therapeutic range and requires individualisation of dosage regimen by maintaining a certain plasma or blood concentration within a targeted therapeutic window | R | 5 | 0.5 | 95.8 |
| | | C | 5 | 1 | 87.5 |
| J | **JKUT (Drug Review & Therapeutic Committee) Quota Medication**—Patient requiring JKUT medication(s) HKL that has limited quota. | R | 3.5 | 1.25 | 50 |
| | | C | 4 | 1 | 54.1 |
| A | **Anaesthetic Referral**—Disease state of a patient requiring ventilator support and referral to the anaesthetist | R | 5 | 1 | 83.3 |
| | | C | 4 | 1.25 | 75 |
| M | **Medication-related Admission**—Current admission is associated with drug related problems such as drug allergy and adverse drug events as well as toxic or poisoning cases. | R | 5 | 0 | 100 |
| | | C | 5 | 0.25 | 95.8 |
| S | **Specialty Care**—Disease state of a patient requiring referral to other specialty care for their expert managements Patient under certain specialty care for their expert management requiring pharmaceutical monitoring | R | 5 | 1 | 87.5 |
| | | C | 5 | 1 | 79.2 |

justified the low consensus in its relevance to ward pharmacy practice. Therefore, this criteria was removed from PAST in round 1.

The completeness for "Anaesthetic referral" was at the lower limit of 75%, which prompted us to modify the content according to experts' opinions. It was then rephrased into "Intensive Care transition" to include patients transferred in and out from critical care ward who require higher pharmaceutical attention.

For the scoring system and PAL level in PAST, 12 (86%) experts agreed whereas 2 experts (14%) did not agree on the scoring system and the corresponding acuity level. A few experts mentioned that the threshold of "score 4 and above" to categorize patient as "PAL level 2— High priority of clerking" was too low. They suggested to increase the scoring threshold for patient to be categorized as PAL level 1 and 2. The scoring and corresponding was then modified in PAST.

Overall, experts comments collected from Delphi Round 1 were substantial. Experts have also recommended a number of potentially important indicators for prioritising patients in medical wards. For example, experts suggested non-high alert medications such as "UKK items" ("*Ubat Kelulusan Khas*"—items that require special approval from director of Ministry of health for its use), drugs that require close monitoring and some patient-related factors such as obesity to be included in PAST tool.

## Delphi round 2

All the new indicators suggested by experts in Round 1 were added into PAST for evaluation in Round 2. As some indicators can be grouped under the same theme, these indicators were reorganized into 8 main criteria, of which some criteria will have sub-components. The new criteria that was added into PAST was 'Specific drugs for close monitoring', 'Medications-related issues', and 'Patient-related factors' (Table 3). Considering that certain criteria contains sub-components, experts were required to rate in detail the overall relevance and completeness for the main criteria statement, and the relevance of each sub-component.

A total of 23/24 (95.8%) experts responded in Delphi round 2. All 8 main criteria statement achieved consensus for both relevance and completeness in Delphi round 2. Most criteria and sub-components achieve consensus with a median score of 4–5 (IQR = 0–1). Only 3 out of 32 sub-components did not achieve consensus for relevance and was removed from PAST. The 3 subcomponents removed from PAST are: "Brain" and "Bone marrow" under criteria Organ dysfunctions, and "Gastroenterology" under Specialty care referral. For "Special populations" under "Patient related factor", 2 experts suggested adding "Immunocompromised patients", which were then added under the sub-component "Special populations".

Regarding the scoring system of PAST, 20/24 experts (87%) agreed on the current scoring system and corresponding PAL level. No modification was done to the scoring system as consensus is considered achieved.

## Delphi round 3

In Delphi round 3, the 8 main criteria and 28 subcomponents retained from Delphi round 2 were presented back to the experts. All 24 experts (100%) responded in Delphi round 3.

Criteria "High Alert Medications" and "Critical Care Transitions" were not open for rating in Delphi round 3 as there were no modifications on the content or structure of the criteria. On the other hand, criteria that has achieved consensus in second round, but have undergone minor changes in the content and sentence structure of the main statement or its subcomponents, were opened for rating again by the experts in third round. For example, footnote was added for "Organ Dysfunction" for further explanation of its definition for a better understanding of the

**Table 3. Results from Delphi round 2 and 3.**

| | | | Round 2 | | | Round 3 | | |
|---|---|---|---|---|---|---|---|---|
| | Experts response rate (%) | | 23/24 (95.8%) | | | 24/24 (100%) | | |
| | Criteria | | Median | IQR | Consensus % | Median | IQR | Consensus % |
| S | **Specific Drugs for Close Monitoring** Drugs that require close monitoring for their: I. Side effects/adverse events II. Infusion-related reaction III. Therapeutic efficacy | R | 5 | 1 | 91.3 | 5 | 1 | 100 |
| | □ Antibiotics listed in Antibiotic Request Form (ARF) | R | 5 | 0.5 | 91.3 | 5 | 0 | 91.6 |
| | □ Anti-tuberculosis drugs | R | 5 | 1 | 86.9 | 5 | 0.75 | 91.6 |
| | □ Biologics (e.g. Adalimumab, Infliximab) | R | 5 | 0 | 86.9 | 5 | 0 | 95.8 |
| | □ Cosmofer®/Venofer® | R | 5 | 1 | 82.6 | 5 | 0.75 | 95.8 |
| | □ Immunoglobulin (IVIG) | R | 5 | 1 | 91.3 | 5 | 1 | 91.6 |
| | □ Prothrombin Complex Concentrate | R | 5 | 1 | 86.9 | 5 | 0.75 | 91.6 |
| | □ Ubat Kelulusan Khas (UKK) drugs | R | 5 | 1 | 82.6 | 4 | 1 | 83.3 |
| | *Overall completeness* | C | 5 | 1 | 86.9 | 5 | 1 | 87.5 |
| T | **Therapeutic Drug Monitoring (TDM)** Patient on medication(s) that requires Therapeutic Drug Monitoring (TDM) • Refer Appendix 2 –TDM list | R | 5 | 0.5 | 100 | 5 | 0 | 95.8 |
| | | C | 5 | 1 | 91.3 | 5 | 0.75 | 95.8 |
| O | **Organ Dysfunction (s)** Patient with one OR more decompensated organ function(s) requiring pharmaceutical intervention | R | 5 | 0.5 | 87.0 | 5 | 0 | 95.8 |
| | □ Brain | R | 4 | 2 | 69.6* | - | - | - |
| | □ Heart | R | 5 | 1 | 87.0 | 5 | 1 | 91.6 |
| | □ Kidney | R | 5 | 0 | 95.7 | 5 | 0 | 95.8 |
| | □ Liver | R | 5 | 0 | 95.7 | 5 | 1 | 95.8 |
| | □ Lung | R | 5 | 1 | 78.3 | 5 | 1 | 83.3 |
| | □ Bone Marrow | R | 4 | 1.5 | 73.9* | - | - | - |
| | *Overall completeness* | C | 5 | 1 | 82.6 | 5 | 1 | 95.8 |
| R | **Specialty Care Referral** Patient under certain specialty care for their expert management requiring pharmaceutical monitoring | | 5 | 1 | 86.9 | 5 | 1 | 87.5 |
| | □ Gastroenterology | R | 4 | 1.5 | 73.9* | - | - | - |
| | □ Geriatric | R | 5 | 1 | 87.0 | 5 | 1 | 87.5 |
| | □ Haematology | R | 5 | 1 | 86.9 | 5 | 1 | 83.3 |
| | □ Infectious Disease | R | 5 | 1 | 91.3 | 5 | 1 | 87.5 |
| | □ Palliative Care / Acute Pain Services | R | 5 | 1 | 87.0 | 5 | 1 | 95.8 |
| | □ Rheumatology | R | 5 | 1 | 82.6 | 5 | 1 | 83.3 |
| | *Overall completeness* | C | 4 | 1 | 86.9 | 5 | 1 | 83.3 |
| I | **Intensive / Critical Care Transition** Patient requiring referral to OR transition of care from critical care team | R | 5 | 1 | 100 | -† | - | - |
| | | C | 5 | 1 | 91.3 | - | - | - |
| M | **Medication-Related Issues** (Main statement) Drug related issues leading to hospitalization OR identified during ward stay | R | 5 | 0 | 95.6 | 5 | 1 | 91.6 |
| | □ Administration (e.g. Nasogastric Tube) | R | 5 | 1 | 82.6 | 5 | 1 | 87.5 |
| | □ Adverse Drug Event(s) | R | 5 | 0 | 95.6 | 5 | 0.75 | 95.8 |
| | □ Device-Related | R | 4 | 1 | 86.9 | 5 | 1 | 95.8 |
| | □ Drug-Drug Interaction(s) | R | 5 | 0.5 | 95.6 | 5 | 1 | 91.6 |
| | □ Medication Error(s) | R | 5 | 1 | 91.3 | 5 | 0 | 95.8 |
| | □ Non-Adherence | R | 5 | 1 | 95.6 | 5 | 1 | 91.6 |
| | □ Polypharmacy ($\geq$ 5 drugs) | R | 5 | 0.5 | 100 | 5 | 1 | 91.6 |
| | □ Underdose / Overdose | R | 5 | 0 | 100 | 5 | 1 | 91.6 |
| | *Overall completeness* | C | 5 | 1 | 86.9 | 5 | 1 | 95.8 |

(*Continued*)

**Table 3.** (Continued)

| | | | Round 2 | | | Round 3 | | |
|---|---|---|---|---|---|---|---|---|
| | **Experts response rate (%)** | | **23/24 (95.8%)** | | | **24/24 (100%)** | | |
| | **Criteria** | | **Median** | **IQR** | **Consensus %** | **Median** | **IQR** | **Consensus %** |
| A | **High Alert Medications** Patient on High Alert Medications (HAM) for <u>more than 24 hours</u> Refer Appendix 1: **Modified HAM List** | R | 4 | 1 | 95.7 | - [†] | - | - |
| | | C | 5 | 1 | 87 | - | - | - |
| P | **Patient Related Factor** | R | 5 | 0.5 | 91.3 | 5 | 1 | 83.3 |
| | □ History of Adverse Drug Reactions (ADR)/Allergy | R | 5 | 1 | 91.3 | 5 | 1 | 87.5 |
| | □ Multiple admissions ($\geq$ 3 within 6 months) | R | 5 | 1 | 87 | 5 | 1 | 87.5 |
| | □ History of falls within 3 months | R | 5 | 1 | 78.2 | 5 | 2 | 66.7[#] |
| | □ Special populations: • Obesity / Underweight • Pregnancy • Immunocompromised | R R R | 5 5 - | 1 | 86.9 | 5 5 5 | 1 1 1 | 87.5z 79.2 95.9 |
| | *Overall completeness* | C | 5 | 1 | 91.3 | 5 | 1 | 83.3 |

*Criteria or sub-components that did not achieve consensus in Round 2 and hence removed from PAST

# Sub-component that did not achieve consensus in Round 3 and removed from final PAST

[†] Criteria that have achieved consensus with no changes were not opened for rating in Round 3

R = Relevance, C = Completeness

experts. We have also added an option of "Others" for the criteria "Specific Drugs for Close Monitoring" and "Specialty Care Referral" based on experts feedback in Round 2.

The results from Delphi round 3 were similar to Round 2, with a slight increase in the consensus in majority of the criteria and subcomponents (Table 3). However, in this round, the relevance of "History of fall" under "Patient Related Factor" has dropped from 78.2% in Round 2 to 66.7%, hence did not pass the consensus benchmark. Some experts suggested to rephrase the criteria into "Falls due to drugs", as not all falls are drug-related. This criteria was not amended in the final Delphi round and hence removed from PAST. All 8 criteria and 29 sub-components that have achieved consensus were retained in the final PAST. Majority of experts (N = 23, 95.8%) agreed on the PAST scoring and the corresponding PAL level in Round 3.

The experts rating on PAST scoring system from all Delphi rounds are summarized in Table 4 and the final scoring system in the PAST is shown in Table 5. The criteria in our final PAST were then rearranged to form a mnemonic 'STORIMAP' as the final pharmaceutical assessment screening tool developed from this study.

## Discussion

The roles of clinical pharmacist are often restrained by short staffing, high patient load and scarcity of resources especially in developing countries like Malaysia. Thus, it is crucial for

**Table 4. Rating results of PAST scoring system.**

| | **Score** | **Patient Acuity Level (PAL)** | **Clerking Priority** |
|---|---|---|---|
| □ | 7 and above | Level 2 | High priority–Full Clerking (**CP2 Part A, B, C, D**) |
| □ | 2–6 | Level 1 | Low priority–Quick clerking (**CP2 Part A, C & D**) |
| □ | 0–1 | Level 0 | Pharmacotherapy review |

**Table 5. Modified scoring system, PAL, and clerking priority in PAST tool from Delphi round 2 and 3.**

| Scoring System | Round 1 (%) n = 14 | Round 2 (%) n = 23 | Round 3 (%) n = 24 |
|---|---|---|---|
| Agree | 12 (86%) | 20 (87%) | 23 (95.8%) |
| Disagree | 2 (14%) | 3 (13%) | 1 (4.2%) |

clinical pharmacists to prioritise patients that require intensive and prompt pharmaceutical care to ensure safe and efficient delivery of clinical pharmacy services [15]. Pharmaceutical assessment screening tools (PAST) address this issue by assisting clinical pharmacists to quickly identify the important factors for prioritising patient care. A United Kingdom pilot study in 2018 reported that screening the patient acuity level (PAL) with a PAST can save pharmacists up to an hour per 25-bed ward per day by not seeing patients with the lowest PAL every day [16].

STORIMAP, a mnemonic name of PAST developed in this study may act as a guide for pharmacists to apply their expert judgement to identify patients with the highest acuity or complexity who require immediate attention by pharmacist. STORIMAP was developed through 'brainstorming' sessions among its group members consisting of senior clinical pharmacists and validated by anonymous panels of experts using three rounds of Delphi survey. Delphi method is an ideal platform where expression and exchange of expert's opinion can be quantified and measured [17].

STORIMAP covers many aspects that a clinical pharmacist should consider when screening patients in the medical wards, that is, for example patient- and medication-related factors, patient co-morbidities, and involvement of any other specialty care. By prioritising patient with higher PAL, pharmacist could identify drug-related issues quickly and thus interventions could be addressed early to optimise patient outcome [6]. The STORIMAP, like other published PAST, may also build up the confidence of clinical pharmacists to quickly stratify the PAL and thus to monitor patients based on their pharmaceutical care needs [18].

The initial PAST in pilot Delphi study had only 7 criteria with no sub-components. From Delphi survey round 1, 6 criteria had achieved consensus while 1 criteria, namely "*JKUT* Quota Medications" was removed. In HKL, the *JKUT* drug list was primarily established for medications that are costly. However, this indicator did not achieve consensus as experts commented that the "*JKUT drugs list were confounded largely by cost of certain medications*" and does not necessitate close monitoring or correlate with patient clinical outcome. In Malaysia, medication expenses is subsidised by the government. Thus, certain measures are needed for cost effective use of medicines to relieve the financial burden. In most studies, the use of expensive medications is not included as part of the criteria in the screening tools (5, 7, and 19). Nevertheless, by utilising STORIMAP, pharmacists can possibly make significant cost saving in healthcare budget through cost avoidance such as preventing DRP and reducing adverse drug events that may lead to prolonged hospitalisation and severe clinical conditions.

Furthermore, indirect cost saving may also be obtained through implementation of STORIMAP in terms of discontinuing unnecessary medicines, switching to effective but less expensive drugs, advocating intravenous to cheaper and safer oral medications and others. Hence, despite expert consensus to remove "JKUT" criteria from STORIMAP, pharmacists may still apply cost effective approach in their daily routine when utilising STORIMAP in screening patients.

Additional criteria of non-high alert medications such as "UKK drugs" were added to STORIMAP at Delphi round 2. The "UKK drugs" (UKK = *Ubat kelulusan khas*) refers to medications that are either not listed in the Malaysian drug formulary, or being used for different indications (off-label use) than the one approved by the Drug Control Authority Malaysia.

Hence, the use of these drugs should be closely monitored and added to STORIMAP as sub-component of "Specific drugs for close monitoring".

Another important subcomponent in STORIMAP is antibiotics with restricted use and/or under the surveillance of antimicrobial stewardship (referred to as "Antibiotics listed in antibiotics request form (ARF)"). Ever since the discovery and development of penicillin, antimicrobials play a huge part in modern medicine especially in decreasing morbidity and mortality due to infectious diseases [19]. However, in recent years, the rise in antimicrobial resistance is a challenge associated with high numbers of death due to improper use of antimicrobials. In 2018, Alvarez-Uria G *et al.* estimated that third generation cephalosporins and carbapenems could be ineffective against a considerable proportion of infections by *Escherichia coli* and *Klebsiella pneumoniae* by 2030 [20]. The cause of antimicrobial resistance is multifactorial, and these include inappropriate choices of antibiotics, inappropriate dose, frequency, and duration. Clinical pharmacists can play a major role in quickly identifying these issues and preventing the increase in the rate of antimicrobial resistance by making necessary interventions. This will then assist the hospital antimicrobial stewardship team towards promoting judicious use of antibiotics [21]. The opinions of our experts to include this indicator in the tool was vital in the era of antimicrobial resistance. Similar with other published studies which also included criteria of antibiotics with restricted use as one of the important criteria in the screening tool [7, 16, 17].

Therapeutic Drug Monitoring (TDM) involves measuring drug concentration in the blood and interpret the individual drug dosage, taking into account the efficacy and safety elements. Medications that require TDM includes drugs with narrow therapeutic index, drugs with variable pharmacokinetics, medications in which the target concentration is tough to be monitored and drugs known to cause adverse reactions [22]. TDM is listed as one of the criteria in STORIMAP that require pharmacists to prioritise patient care and case clerking. Similar to STORIMAP, fifteen other screening tools also included TDM as their screening criteria [23].

In Australia, the criteria used for case prioritisation among pharmacists include: patients on high-risk medications which consist of anti-infectives, TDM medications, patient's with renal impairment, non-therapeutic INR (international normalized ratio)/aPTT (activated partial thromboplastin time), transfer from high dependency area, elderly patients, deranged electrolytes, sub-therapeutic platelet count (associated with anticoagulant use) and number and type of comorbid conditions [15]. These criteria are similar to those included in STORIMAP. Most of the published prioritisation tools are applicable for general use among hospitalised patients. However, STORIMAP is more specific on medical adult patients which is distinctive from other tools, such as one tool published in Australia which focuses on paediatric patients [24].

Despite STORIMAP was validated as a new and comprehensive tool for screening medical patients, there are a number of limitations in this study. First, STORIMAP was specifically developed for screening adult patients in general medical wards only. However, the basic indicators found in other published tools have also been incorporated in STORIMAP. Therefore, STORIMAP can be potentially extended and applied to patients from other disciplines. Secondly, STORIMAP has only been validated by senior pharmacists. Other healthcare professionals such as medical physicians and nurses were not invited as part of the expert panels. Even so, the tendency of STORIMAP missing out important factors is minimal as it is adopted from a few comprehensive screening tools available [6, 8–10]. Thirdly, the STORIMAP tool could be more time consuming for clinical pharmacists to use in comparison to other simpler tools available. This is due to the extent of details required to be filled in manually before allocating the PAL for each patient. Nonetheless, the comprehensive nature of this tool also

ensures that patients with higher PAL needing more pharmaceutical attention will not be missed.

## Conclusion

Our Delphi study has led to the development of a validated PAST named STORIMAP. This tool is developed to guide clinical pharmacists to efficiently identify high risk patient that needs prompt and intensive pharmacist review during their hectic daily routine. The finalised tool is designed in such a way that is comprehensive, simple to use, and user friendly. Further research is needed to determine the acceptability and applicability of this tool in assisting clinical pharmacists to prioritise patient care in medical wards.

## Supporting information

**S1 File. Minimal data set.**
(DOCX)

## Acknowledgments

We would like to thank the Director-General of Health Malaysia for his permission to publish the research findings. We would also like to thank Ms Noraini Mohamad, the Deputy Director (Pharmacy) of HKL, for giving us support in conducting this study, Ms Rohana Hassan for her guidance, Ms Nirmala a/p Jagan for assisting in the literature research and Clinical Research Centre (CRC) of HKL for continuous support. Lastly, we would like to thank all expert panels for their participation and contribution of constructive opinions to this study.

## Author Contributions

**Conceptualization:** Rahela Ambaras Khan.

**Data curation:** Cheok Ee Chang, Chan Yen Tay, Baavaanii Thangaiyah, Victor Sheng Teck Ong, Sabariah Pakeer Oothuman, Shazwani Zulkifli, Nur Fatin Najwa Azemi, Pavithira Subramaniam.

**Formal analysis:** Cheok Ee Chang, Chan Yen Tay, Baavaanii Thangaiyah, Victor Sheng Teck Ong, Sabariah Pakeer Oothuman, Shazwani Zulkifli, Nur Fatin Najwa Azemi, Pavithira Subramaniam.

**Funding acquisition:** Rahela Ambaras Khan.

**Investigation:** Cheok Ee Chang, Chan Yen Tay, Baavaanii Thangaiyah, Victor Sheng Teck Ong, Sabariah Pakeer Oothuman, Shazwani Zulkifli, Nur Fatin Najwa Azemi.

**Methodology:** Cheok Ee Chang, Baavaanii Thangaiyah, Victor Sheng Teck Ong, Sabariah Pakeer Oothuman, Shazwani Zulkifli, Nur Fatin Najwa Azemi.

**Project administration:** Cheok Ee Chang, Rahela Ambaras Khan, Chan Yen Tay.

**Supervision:** Cheok Ee Chang, Rahela Ambaras Khan, Chan Yen Tay.

**Visualization:** Rahela Ambaras Khan.

**Writing – original draft:** Cheok Ee Chang, Baavaanii Thangaiyah, Victor Sheng Teck Ong, Sabariah Pakeer Oothuman, Shazwani Zulkifli.

**Writing – review & editing:** Cheok Ee Chang, Rahela Ambaras Khan, Chan Yen Tay.

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
