## [Decision Letter · Decision Letter 0]

28 Nov 2022

PONE-D-22-28558Development and validation of a pharmaceutical assessment screening tool to prioritise patient care in a tertiary care hospitalPLOS ONE

Dear Dr Cheok Ee Chang

Thank you for submitting your manuscript to PLOS ONE. After careful consideration, we feel that it has merit but does not fully meet PLOS ONE’s publication criteria as it currently stands. Therefore, we invite you to submit a revised version of the manuscript that addresses the points raised during the review process.

We look forward to receiving your revised manuscript.

Kind regards,

Muhammad Junaid Farrukh

Academic Editor

PLOS ONE

Journal Requirements:

Reviewers' comments:

Reviewer's Responses to Questions

**Comments to the Author**

1. Is the manuscript technically sound, and do the data support the conclusions?

Reviewer #1: Yes

Reviewer #2: Yes

2. Has the statistical analysis been performed appropriately and rigorously? 

Reviewer #1: Yes

Reviewer #2: No

3. Have the authors made all data underlying the findings in their manuscript fully available?

Reviewer #1: Yes

Reviewer #2: Yes

4. Is the manuscript presented in an intelligible fashion and written in standard English?

Reviewer #1: Yes

Reviewer #2: Yes

5. Review Comments to the Author

Reviewer #1: The study “Development and validation of a pharmaceutical assessment screening tool to prioritise patient care in a tertiary care hospital” is well-conducted and reported. I have a few minor comments, which I think will make the reporting slightly better.

1. I could not find how the Patient acuity level scoring was performed. What weights were given to each criterion and/or sub-component. I am not sure if it is 1/0 scoring or some other weights for the 8 criteria, unless I overlooked it. I would recommend authors to add a sentence or two describing the scoring method that reached the consensus.

2. Were the experts involved in the pilot Delphi round also part of the Delphi surveys 1–3? I would recommend the authors to add a sentence regarding this in the methods.

3. A typo: in table 2 first row, “High Alert Medication” is written twice.

Reviewer #2: The manuscript presented an interesting introduction to the purpose of the study. However, the tools researchers developed for prioritizing patients for intensive pharmaceutical care is not new and has mostly adapted from published literature.

Most of the expert panel members were female (83.3%), Is it similar to the proportion of clinical pharmacist in the three selected hospitals to be a representative data?

Provide expansion of abbreviations used in table 2 in table legend.

Again, all abbreviations are not expanded in table 3 legend.

The tool developed as PAST is presented merely in percentage and grading for consensus, how reliable this tool in the absence of reliability testing is not clear to me.

The sample size is too small to come to the conclusion of the reliability of the outcome.

I hope all participants who are said to have experience in clinical pharmacy unit shall also by default will have experience of working in pharmacy as clinical pharmacy unit is within pharmacy. However, the demographics is confusing in table 1. I can find different levels of experience shown in this two categories. 9-10 years’ experience with only 6 participants while there is no 9-10 years’ data in clinical pharmacy, it jumps from 8-9 and then more than or equal to 10 years, there is no data of 9-10 years.

What is distribution of participants from different centers to understand the diversification of them.

The first three paragraphs of discussion are more less repeat of introduction; it needs to discuss the results with interpretation in consideration with other published work.

How they ensured anonymity of the participants in this study especially if was done face to face, I hope atleast a part of it was done face to face.

How this tool which author refer as STORIMAP is considered as validated? Authors claim in line 409.

Do they have criteria, content and construct validity checked? Did they do reliability check?

6. PLOS authors have the option to publish the peer review history of their article (what does this mean?). If published, this will include your full peer review and any attached files.

Reviewer #1: No

Reviewer #2: No

---

## [Author Response · Author response to Decision Letter 0]

6 Feb 2023

Thank you for inviting us to submit a revised draft of our entitled ‘Development and validation of a pharmaceutical assessment screening tool to prioritise patient care in a tertiary care hospital’ for publication as a research article in the PLOS ONE journal. We also appreciate the time and effort you and each reviewer have dedicated to providing insightful opinions on ways to strengthen our paper. Thus, it is with great pleasure that we resubmit our article for further consideration. We have incorporated changes that reflect the detailed suggestions you have graciously provided. 

To facilitate your review of our revisions, we attached a document for the point-by-point response to the questions and comments delivered in your email dated 29 November 2022. 

Again, thank you for allowing us to strengthen our manuscript with your valuable comments and queries. We look forward to hearing from you regarding our submission. We would be glad to respond to any further questions and comments that you may have. 

We hope that the revisions in the manuscript and our accompanying response will be sufficient to make our manuscript suitable for publication in the PLOS ONE journal.

Thank you for your time and attention.

---

## [Editor Report · Decision Letter 1]

14 Feb 2023

Development and validation of a pharmaceutical assessment screening tool to prioritise patient care in a tertiary care hospital

PONE-D-22-28558R1

Dear Cheok Ee Chang,

We’re pleased to inform you that your manuscript has been judged scientifically suitable for publication and will be formally accepted for publication once it meets all outstanding technical requirements.

Kind regards,

Muhammad Junaid Farrukh

Academic Editor

PLOS ONE

Additional Editor Comments (optional):

Reviewers' comments:

<quillbot-extension-portal></quillbot-extension-portal>

---

## [Editor Report · Acceptance letter]

22 Feb 2023

PONE-D-22-28558R1 

Development and validation of a pharmaceutical assessment screening tool to prioritise patient care in a tertiary care hospital 

Dear Dr. Chang:

I'm pleased to inform you that your manuscript has been deemed suitable for publication in PLOS ONE. Congratulations! Your manuscript is now with our production department. 

Kind regards, 

on behalf of

Dr. Muhammad Junaid Farrukh 

Academic Editor

PLOS ONE